# CANAttack: Assessing Vulnerabilities within Controller Area Network

**DOI:** 10.3390/s23198223

**Published:** 2023-10-02

**Authors:** Damilola Oladimeji, Amar Rasheed, Cihan Varol, Mohamed Baza, Hani Alshahrani, Abdullah Baz

**Affiliations:** 1Department of Computer Science, Sam Houston State University, Huntsville, TX 77340, USA; dko011@shsu.edu (D.O.); axr249@shsu.edu (A.R.); cxv007@shsu.edu (C.V.); 2Department of Computer Science, College of Charleston, Charleston, SC 29424, USA; 3Department of Computer Science, College of Computer Science and Information Systems, Najran University, Najran 61441, Saudi Arabia; hmalshahrani@nu.edu.sa; 4Department of Computer Engineering, College of Computer and Information Systems, Umm Al-Qura University, Makkah 21955, Saudi Arabia; aobaz01@uqu.edu.sa

**Keywords:** Controller Area Network (CAN), autonomous systems, intrusion, security, vulnerability andthreat modeling

## Abstract

Current vehicles include electronic features that provide ease and convenience to drivers. These electronic features or nodes rely on in-vehicle communication protocols to ensure functionality. One of the most-widely adopted in-vehicle protocols on the market today is the Controller Area Network, popularly referred to as the CAN bus. The CAN bus is utilized in various modern, sophisticated vehicles. However, as the sophistication levels of vehicles continue to increase, we now see a high rise in attacks against them. These attacks range from simple to more-complex variants, which could have detrimental effects when carried out successfully. Therefore, there is a need to carry out an assessment of the security vulnerabilities that could be exploited within the CAN bus. In this research, we conducted a security vulnerability analysis on the CAN bus protocol by proposing an attack scenario on a CAN bus simulation that exploits the arbitration feature extensively. This feature determines which message is sent via the bus in the event that two or more nodes attempt to send a message at the same time. It achieves this by prioritizing messages with lower identifiers. Our analysis revealed that an attacker can spoof a message ID to gain high priority, continuously injecting messages with the spoofed ID. As a result, this prevents the transmission of legitimate messages, impacting the vehicle’s operations. We identified significant risks in the CAN protocol, including spoofing, injection, and Denial of Service. Furthermore, we examined the latency of the CAN-enabled system under attack, finding that the compromised node (the attacker’s device) consistently achieved the lowest latency due to message arbitration. This demonstrates the potential for an attacker to take control of the bus, injecting messages without contention, thereby disrupting the normal operations of the vehicle, which could potentially compromise safety.

## 1. Introduction

As the number of electronic features embedded in modern cars continues to increase, cars can now be compared to advanced computers operating with numerous digital nodes [1]. These nodes provide drivers with a wide range of benefits that transcend mere transportation, encompassing enhanced comfort and ease. Examples of these nodes are the Anti-lock Braking System (ABS), the Engine Control Module (ECM), the airbag module, infotainment module, the instrumental cluster panel, and so on. The successful integration of advanced car nodes has been made possible by various features, particularly the in-vehicle communication protocol. In-vehicle communication protocols ensure that the various nodes within the car communicate using standardized methods with each other. Simply put, these protocols allow for the effective exchange of information and data between the different electronic nodes when the car is in operation [2,3,4,5]. One of the highly prevalent in-vehicle protocols used in modern cars is the Controller Area Network (CAN) bus [6]. Most car vendors adopt this communication protocol because it offers reliability and real-time capabilities with minimal power consumption [7]. The technical terms for nodes in a CAN-enabled system are Electronic Control Units (ECUs). As earlier stated, these nodes or ECUs refer to the various electronic systems effectively communicating with each other via the CAN bus in the car.

The automobile industry has witnessed technological growth over the years, with continuous improvements in the vehicles being produced yearly. A notable improvement in this industry is the invention of autonomous vehicles or self-driving cars [8]. Famous automobile companies that produce these high-performance cars are Tesla, Waymo, General Motors (GM), and Ford. It is worth mentioning that these famous automakers employ the CAN bus in the manufacturing of self-driving cars to facilitate the required integration and coordination functionalities amongst the various ECUs, primarily due to the numerous benefits it offers, such as increased reliability, improved efficiency, resilience, and cost-effectiveness.

Although the CAN bus was initially introduced in the early 1980s [9], its significance in today’s technological era cannot be overemphasized, especially as automakers continue to adopt it in deploying increasingly sophisticated vehicles. However, a major area of concern arises when it comes to security. The CAN bus was not initially designed with security as a focal point [10]; instead, its primary focus was on reliability in communication while using minimal resources such as wires and power consumption. This design philosophy facilitated the integration of ECUs into vehicles, increasing their overall sophistication. As a result, as cars became more sophisticated, the frequency of attacks against them increased. Therefore, there is an imperative need to investigate the security vulnerabilities associated with the CAN bus.

Despite the numerous benefits of the CAN, the issue of security is of utmost importance, as the adverse effects of an attack on a CAN-enabled vehicle can be life-threatening to passengers and cause financial liabilities to car manufacturers [11]. A common security flaw highlighted by most researchers within the CAN bus is the lack of encryption in the messages exchanged between the various ECUs [12,13]. That is, the messages are sent in plain text. For instance, attackers can hack into the car to tamper with the ECUs (such as the ABS and/or airbags), intercept the messages, and alter them to make them ineffective when needed [14].

In this paper, we aimed to extend our work in [1], where we detailed the steps to create a testbed for a CAN-enabled system with four ECUs attached to it. The testbed created in this work will be the base system for investigating the security vulnerabilities within the CAN bus. We discuss the proposed attack scenario and implemented this attack scenario in three cycles to compare the effect of the attack as time increases. Furthermore, we conducted a comparative analysis to monitor the system’s performance during normal and attack operations. Additionally, we studied the latency within the system to monitor the time delay between messages transmitted via the bus. The result of the assessment revealed that the CAN arbitration feature could be used as an attack vector by an attacker to flood the bus with irrelevant messages after gaining access to the bus. This attack prevents the effective transmission of legitimate messages from other nodes within the vehicle, thereby disrupting its general operations. Furthermore, the examination of the latency distribution of the vehicle under attack in the proposed scenario shows the extensive control that the attacker has over the vehicle in transmitting messages. We recorded relatively low latency from the compromised node (attacker’s device) in all scenarios monitored and high latency for legitimate messages from other nodes on the bus. This shows that the attacker is able to send messages rapidly with little competition from other nodes due to the malicious messages consistently winning the arbitration process.

The rest of this paper follows the following structure: Section 2 gives an overview of the CAN bus protocol and existing relevant works. Section 3 discusses the testbed used as the base system for security assessment and the threat model identified in this research. Section 4 discusses the security vulnerability in this research, explaining the proposed attack design, the implementation cycles, and the simulations. Section 5 explains the experiments, the results, and the latency of the system under attack. Finally, Section 6 explains the conclusion, limitations, and future works.

## 2. Background and Related Work

In this section, we give a brief overview of the CAN bus protocol, highlighting its structure, communication mechanism, and message format. Additionally, we examine existing literature that tackles attacks on the CAN bus system over the years.

### 2.1. CAN Bus Overview

A thorough grasp of the CAN bus protocol is required to develop a standardized CAN bus system for cars. Here, we summarize the fundamental components of the CAN bus protocol. We focus on the bus architecture and data transmission mechanisms within the bus.

Bosch [9] created the CAN bus in 1983 as a multi-master message broadcast system. It was created to communicate at a maximum signaling rate of 1 Mbps. Unlike traditional networks such as USB or Ethernet, CAN does not use the same protocol to send huge data blocks from one node to another. It is a serial communication bus created for the vehicle industry by the International Standardization Organization (ISO). Its major goal was to eliminate complicated wiring harnesses in favor of a simpler two-wire bus arrangement. The CAN bus comprises a controller and a transceiver responsible for transmitting and receiving information among subsystems [15]. It incorporates object and transfer layers for message filtering and status handling.

Communication within the CAN bus occurs through different pair signals: CAN high and CAN low. The standard data rates for the configuration include 125 kbps, 500 kbps, and 1 Mbps. In the latest CAN bus standard, the CAN frame allows transfers of up to 64 B at higher speeds [16]. Figure 1 illustrates a typical physical configuration of a CAN network with an arbitrary number of nodes. These nodes communicate via the two-wire CAN bus protocol, where the lines are CAN high and CAN low.

The CAN protocol is a broadcast network; as a result, all ECUs connected to the bus can pick up signals/messages sent via the bus. Signals sent on the bus are referred to as frames. These frames consist of the messages alerting the vehicle (or system) of the operations to be performed. The length of CAN messages can be in two formats: the standard format and the extended format. The extended CAN message format differs from the standard CAN format slightly due to the inclusion of the additional 18 bit in the arbitration field [17]. Figure 2 shows the frame format of an extended CAN message.

The fields in Figure 2 are briefly described below:SOF: This is the “Start of Frame”. This denotes the beginning of the message.Arbitration: This field consists of five components in the extended frame format.Identifier: This subfield decides which message takes precedence in the standard frame format.SRR: This stands for “Substitute Remote Request”. It contains 1 bit, and it is always recessive. It signifies that the bus uses data frames instead of remote request frames.IDE: This is the “Identifier Extension” sub-field. It signifies the frame format. The standard frame format is dominant “0”, and extended is represented as “1”.Identifier: These additional 18 bit are used to signify the arbitration on extended frame formats.RTR: This represents “Remote Transmission Request”. This sub-field is used to indicate if the frame is a request for data or the transmission of data.Control: This is used to represent the data length on the bus.Data: This is the actual message on the bus.CRC: This represents the “Cyclic Redundancy Check”. It checks for errors within the frame.ACK: The ACK sub-field is the “Acknowledgement”. It checks for the message’s reliability and integrity on the bus.EOF: EOF stands for “End of Frame”. It is always recessive (0).

### 2.2. Related Work

Here, we discuss existing literature highlighting security vulnerabilities within the CAN bus. Following high-profile hacking attacks that rattled the automobile industry in recent years, the CAN bus and its security have taken center stage [18,19,20,21,22,23,24,25]. Nodes such as the airbags and WiFi nodes have become the most-prevalent entry points for automotive hackers [26]. For example, the CAN bus can easily be accessed via the Onboard Diagnostics (OBD) port [27], which is also used to obtain vehicle data at service facilities. As a result, the issue of security within the CAN bus has been extensively researched, specifically to identify possible avenues that attackers could easily exploit within a vehicle operating with the CAN protocol. In light of this, we examined these prior works and state our contributions.

The earliest reported attack on the CAN bus system was carried out on the simulation of a car by tampering with the electric window lift by Hoppe and Dittman in 2007 [28]. Following this attack, Kosher et al. [29] carried out various successful attack attempts on other electronic systems within the two CAN-enabled automobiles, particularly the radio, instrumental panel cluster, body controller, engine, and brakes. Their experimental security assessment of the CAN protocol identified vulnerabilities such as Denial-of-Service attacks (DoS), packet sniffing, and fuzzing attacks.

Upon reviewing multiple successful attack mechanisms, it becomes evident that attacking a CAN-enabled system effectively requires targeting a vulnerable node. Researchers commonly agree that such an attack must be conducted by physical or remote access means [11,30]. Physical access attacks involve establishing a direct or indirect connection to the targeted vehicle. In other words, the attacker must create opportunities to connect with the vulnerable node while executing the attack physically. In contrast, remote access attacks are centered on gaining unauthorized entry to the vehicle via wireless means. Current vehicles incorporate various ECUs that facilitate wireless communication, such as Bluetooth, WiFi, radio, telematics, tracking tools, and more [31]. These ECUs can be targeted by the attacker even without being in the same vicinity as the vehicle.

A common physical attack mechanism categorized as direct access is the attack on the OBD port. This port was primarily created by automakers for the diagnostics analysis of the vehicle in real-time. Kosher et al. [29] attacked the OBD-II port to inject fake packets into the CAN bus. The messages sent via the port enabled them to access the brake, preventing it from halting the car while running at 40 mph. Zhang et al. [32] used the OBD-II port to sniff packets sent via the CAN; their attack mechanism involved exploiting the arbitration feature in the CAN to cause DoS attacks within the vehicle.

Currently, remote access attacks are becoming more common due to the impracticality of physical access attacks in real-life scenarios [11]. Checkoway et al. [33] targeted various electronic systems within the vehicle via wireless access. They exploited vulnerable nodes such as CD players, Bluetooth, and radios, concluding that a car door can be unlocked via remote access. Payne [34] remotely hacked into the simulation of a car using a hacking tool that they have now made open access. This tool allowed them to capture packets and send signals that made the vehicle inaccessible for hours. In the scenario centered on the simulation of a car, they accessed the car remotely to cause replay attacks.

Some researchers have focused on attacking the CAN bus via its inbuilt fault tolerance feature for error handling, famously known as the bus-off attack. The attack mechanism revolves around triggering the deactivation of a node upon detecting suspicious activity. Lehira et al. [35] exploited the error-handling mechanism in the CAN bus protocol to launch spoofing attacks against specific ECUs within the vehicle. As a result, regular messages from authorized ECUs were obstructed. Additionally, Bloom [36] proposed an attack mechanism called WeepingCAN. In their research, they attacked a 2016 Kia Optima to identify low-priority signals to create bus-off attacks. Upon examining the feasibility of the WeepingCAN attack in a car, they reported an initial success rate of 75% before detection.

Additionally, Mohammed et al. [37] proposed a novel attack known as physical-layer data manipulation. This attack involves the collaboration of multiple compromised ECUs to induce changes from dominant bits (0) to recessive bits (1) within CAN messages. To execute this manipulation, the attackers generate transient voltages on the CAN bus, capitalizing on the parasitic reactance of the bus and imperfections in the line drivers. They revealed that, when more than eight compromised ECUs coordinate their efforts, they can create a significant voltage drop, resulting in the alteration of dominant bits to recessive ones in transmitted messages.

Prominently, Valasek and Miller [38] conducted a comprehensive survey on 12 car brands and 21 commercial cars to identify remote attack surfaces and assess their difficulty levels. The attack had three stages: compromising the wireless interface ECU, injecting messages to communicate with critical ECUs, and modifying an ECU maliciously. While they anticipated increased vulnerabilities due to growing cyber–physical systems, practical verification was hindered by diverse vehicle applications. Notably, they also remotely hacked a Jeep Cherokee [39], drawing attention to motor vehicle vulnerability, prompting a public announcement.

After analyzing the existing literature, we observed and identified possible means that attackers could exploit, which is the arbitration mechanism. There is a need to examine the impact of the attackers leveraging this means of attack. Therefore, in this paper, we carried out a security assessment of the CAN bus protocol by proposing an attack scenario that exploits the CAN arbitration feature. This feature gives precedence to high-priority IDs, i.e., messages with the lowest ID. Additionally, we monitored the latency of the system under attack to gauge the impact of the attack on the speed and efficiency of message delivery. The work is an extension of the paper [1].

## 3. CAN Bus Testbed and Threats

In this section, we examine the threat model used to identify the vulnerabilities within the system. Furthermore, we highlight the hardware used to build the CAN testbed used as the base system in this research. We outlined the comprehensive steps in developing this testbed in [1].

### 3.1. Threat Model

After analyzing our system under attack and in its normal operations, we can identify three major vulnerabilities. In the event that an attacker uses a compromised node by exploiting the arbitration mechanism, the three attacks we identified are the spoofing attack, injection, and Denial of Service attack:*Spoofing attack*:Spoofing occurs when a compromised node sends CAN data frames with a changed (forged) ID field to masquerade as data or a command from a valid-source ECU node. The spoofing attack is easy to adapt to the CAN bus model. It has adverse effects because it decreases communication performance on the network [40]. Since CAN lacks authentication and the bus is a broadcast network, a compromised ECU might readily deliver CAN frames with any ID, even IDs belonging to other legitimate/critical ECUs. In this research, we achieved this spoofing attack by assuming that, first, an attacker gains physical access to the CAN bus and connects to it, effectively becoming part of the network. Since messages sent via the bus are in plain text, they can easily monitor and understand messages sent and can then skillfully alter bits within the packet frame to manipulate the CAN ID, specifically changing the ID segment to the decimal number 0, which is represented in hexadecimal notation as 0x00.*Injection attack*:Generally, attackers use direct or indirect access points to inject messages into the CAN bus, suppress valid communications (i.e., genuine messages with higher-priority IDs than injected ones are ignored), or penetrate an ECU to perform malicious actions. Attacks against direct access points include the OBD-II port, CD player, and USB port [41]. In our case, we could inject futile messages to the bus via the WiFi node we attached to it. In our attack scenario, we implemented an injection attack where the attacker successfully spoofed the CAN ID to a high-priority ID, in particular 0x00. Consequently, the attacker’s node flooded the bus with fraudulent or irrelevant messages by either continually injecting arbitrary messages into the CAN bus or injecting unauthenticated messages with the spoofed ID into the vehicle. We explain the result of this implementation comprehensively in Section 5.*Denial of Service (DoS) attack*:The CAN protocol is also subject to DoS attacks. CAN’s arbitration system allows higher-priority nodes to talk first. Because of the prioritization on the CAN bus, if a malicious node with the highest priority is always active, the other nodes cannot interact.As shown in this research, an attacker can carry out a DoS attack to render a specific CAN bus system inoperable by conforming to the CAN standard or by breaching it [42]. We achieved this by transmitting as many messages to the CAN bus as physically possible with the smallest feasible ID (0x00). When the bus is idle, if two or more ECUs desire to transmit simultaneously, the one with the lowest ID will have priority (arbitration). As a result, we noticed in the logs that, because the zero ID takes precedence over all other message IDs, none of the normal messages will win the arbitration against the injected message, resulting in the prevention of signal transfer from the regular ECUs.

### 3.2. Hardware

Selecting appropriate hardware components is crucial when constructing a CAN bus system to represent the nodes effectively. In a CAN bus network, all nodes are interconnected in parallel, establishing a direct link between each node and every other node. The necessary hardware to develop the CAN-enabled system is listed below. In Table 1, we highlight the hardware used to develop the physical structure of the system.

### 3.3. Physical Structure of the CAN Testbed

Figure 3 shows the physical layout of the base system used in the security vulnerability assessment in this study.

Once the physical and software components of the node were successfully interconnected to simulate the CAN protocol system, we conducted rigorous testing to ensure its proper functioning. To monitor and analyze the data generated by our system, we used a sniffer called the CL2000. This sniffer allowed us to assess the effectiveness of our system and examine the transmitted data on a host computer. Additionally, we incorporated a protocol decoder, the PicoScope 2204A, into our CAN system. This device effectively parsed messages transmitted on the CAN bus, measuring voltage in relation to time (in milliseconds). In Figure 4, we present a decoded sample message captured by the PicoScope.

## 4. Proposed Attack

In this section, we examine the security vulnerabilities that could be exploited within the bus. We discuss the attack scenario and highlight the steps and hardware used to implement the scenario in our already-existing system. We then discuss the threats identified based on the attack scenario created.

All messages are sent on the bus in plain text. Figure 4 shows the plain message sent within the bus after the PicoScope 2204A decoded it. These signals and messages are generated by the nodes communicating to ensure the optimal performance of the car. Considering that the protocol sends messages without any encryption, it creates an appealing avenue for attacks. Any exploit against a node on the system will disrupt the network if an attacker targets the bus. In light of this lack of security, it is critical to evaluate the security threats to which the CAN bus protocol may be vulnerable.

From a security point of view, a secure network should meet these five measures:*Data integrity*: This means that the message sent was not altered by the time it reached the receiver. The CAN bus has a mechanism called the CRC embedded into the packet to check that the data reach their destination without any changes being made to it.*Authentication*: a process that verifies the validity of a node. The CAN bus does not have a mechanism to perform this check.*Confidentiality*: Only authorized users have access to the communication between sender and receiver.*Non-repudiation*: There should be a way to demonstrate that the parties involved in the communication cannot refute the message’s legitimacy.*Availability*: The system should ensure that message reliability is ensured under all conditions.

To investigate the potential threats to the bus, we devised an attack scenario centered on message injection into the vehicle. As previously stated, the bus uses arbitration, which is a process that determines which message is sent on the bus if a collision occurs. The node that wins the process is always the one with a smaller message ID. Our proposed attack scenario exploits this arbitration process of the CAN bus protocol. This process was also explained using analogies in [1].

### 4.1. Proposed Attack Scenario

We propose to utilize the CAN-enabled system, depicted in Figure 3, to develop our proposed attack scenario. A typical car can have up to one-hundred and twenty-seven nodes connected to the CAN bus system [1]. To simulate the inner mechanism of a real car, we assigned thirty IDs to three of the nodes in the system we developed. As a result, we had a CAN-enabled system working with ninety IDs transmitting messages via the bus.

Recall that the initial system consisted of four ECUs connected to a CAN bus system. Among these ECUs, three were assigned thirty IDs each, while the remaining ECUs served as the compromised node responsible for injecting messages into the bus. In the proposed attack scenario, which is depicted in Figure 5, we aimed to exploit the CAN arbitration mechanism.

In this proposed attack scenario, we simulated over ninety ECUs to communicate on the CAN bus in a car. One of the ECUs was hijacked by an attacker to gain entry into a vehicle and make that vehicle inoperable. The attacker hijacks a node that can be connected and spoofs the IDs of the messages sent to the bus to the smallest number possible (in this case, 0x00). As the attacker begins to send messages, all messages from the other ECUs will be halted because the compromised node has the smallest message ID. Therefore, this means that, when the attacker sends messages at an elevated rate (injection), the only signals transmitted on the bus will be those of the attacker, making other legitimate ECUs inoperable for a particular time (DoS).

Table 2 shows the message IDs assigned to each ECU to carry out the attack in this research. To simulate a vehicle with thirty ECUs communicating via the CAN bus, we assigned ten random IDs each to ECU 1, ECU 2, and ECU 3. We carried out this allocation in the software phase of this research. Using C++, we leveraged <random>, which is a random number generator library. Specifically, we designed a custom script that employed the random number generator functions provided by the <random> library to generate random CAN IDs within the specified CAN ID range. This aided the simulation of ninety ECUs in our implementation. The range of the IDs allocated to each node generated is highlighted in the column called Number of Assigned IDs in Table 2. We also detailed the values of each of these message IDs in hexadecimals (hex).

### 4.2. Metrics Used in the Research

The metrics used are based on the implementation cycles and simulations of the runs. These are explained below:Implementation cycles:We conducted three separate runs for the implementation of the attack scenario. This approach aimed to examine the developing impact of the attack over time. As a result, we compared the system’s performance during normal operation with its performance during the attack operation throughout all three runs to provide a full assessment of the CAN bus operation and its capacity to handle attacks. Table 3 depicts the time for each run during analysis.Simulations:We monitored the traffic of the CAN system from two perspectives (normal simulation and attack simulation) in all three implementation cycles. Table 4 provides a description of these perspectives.

We carried out this examination in all three attack cycles from the two perspectives detailed above (normal and attack) by connecting a sniffer (CL2000) to the CAN system in Figure 3. This sniffer monitored all traffic sent within the system for allotted time limits in each implementation cycle. After the specified time limit had passed, we halted all communication within the system by disconnecting the power supply. We connected the CL2000 to a workstation and retrieved the Comma-Separated Values (CSV) file containing the log of the CAN traffic. From this CSV file, we were able to obtain all message IDs, timestamps, data, and data lengths sent via the bus during a particular implementation cycle. We processed and interpreted these CSV files using Python and discuss the results of our analyses below.

## 5. Experiments and Results

As stated previously, we monitored the system in three different cycles with strict time intervals. Here, we will discuss the result of our evaluation. By monitoring the system during three separate runs, we aimed to compare the effects of the attack over time and gain insights into the resilience and adaptability of the CAN bus under such circumstances. This analysis provides valuable insights for enhancing the security and robustness of automotive systems.

### 5.1. Test Run 1 Results

After running the first attack simulation for 15 min, we observed the traffic from the sniffer and evaluated the data using Python. In this analysis, we counted the occurrence of each message ID in the CSV file containing the observation of the vehicle simulation under attack after 15 min:Attack simulation: After we processed the data using Python, we observed that the compromised node with the ID 0x00 sent the most messages. We recorded the total packets sent from ID 0x00 as 432 packets in the observed time frame in the attack simulation. The next most-active message ID (0x6D) only sent 66 packets, which is an 84.72% decrease compared to the packets sent from the compromised message ID.Normal simulation: We then monitored the system in normal operations, i.e., without the compromised ID 0x00. In this simulation cycle, the message ID that displayed the highest activity (000000D1) transmitted 86 packets, while the following message ID (00000196) sent 82 packets. Our observation of all packets or frames transmitted throughout this simulation revealed an equitable distribution in the sent packets.

We displayed the result of the count of the total number of packets sent by each message ID in the attack and normal simulation, respectively. Table 5 shows the total number of packets sent for the attack simulation, and Table 6 shows the total number of packets sent in the normal simulation in Test Run 1 (15 min).

Subsequently, we visually represent these tabular data in bar charts. Figure 6a gives a visual representation of the packets sent in the test run in the attack scenario, while Figure 6b shows the packets sent during normal scenarios after 15 min. “CAN_ID” in the *x*-axis of the figures represents the message ID, and “count” on the y-axis represents the total number of packets sent by each message ID. In Figure 6a, we see a bar surpassing the other bars in the graph greatly; this bar with the largest count shows the total count for the compromised message ID in the attack scenario. This indicates the DoS attack, clearing emphasizing the control the attacker had to flood the system with packets in 15 min. However, Figure 6b shows the typical distribution of packets when the system operates in its regular mode. This indicates all nodes/ECUs had a fair opportunity to send packets via the bus when needed.

### 5.2. Test Run 2 Results

In the second test run, we monitored the system under normal and attack operations after 30 min. The data were logged using the sniffer discussed prior and saved in CSV files to represent each simulation. Using Python, we processed and interpreted these files, particularly focusing on the message ID column to count the total number of packets sent by each of them. We discuss the results of these occurrences in the two simulations monitored in this cycle below:Attack simulation: In this cycle, the compromised message ID (0x00) sent 3556 packets in 30 min. The following message ID with the second-highest count was 00000130. This message ID could only send 377 packets within 30 min of the attack operation. This shows a percentage difference of 842.72% between the packets sent by the compromised message ID and the legitimate message ID.Normal simulation: Upon analyzing the normal operation, messages were sent in an evenly distributed manner. The highest number of packets sent from a particular message ID was recorded as 106. The message ID that recorded this figure was 00000191, while the subsequent message ID sent 103 packets. Further analysis of the data processed in this phase revealed closely correlated figures in relation to the overall packets sent without any indications of irregular behavior.

Table 7 and Table 8 show the total number of packets sent by the message IDs in the attack and normal simulations in Test Scenario 2, respectively. Furthermore, we visually represent these data using bar charts in Figure 7a,b. Figure 7a shows a visual representation of the total number of packets sent by each ID in the attack simulation. Figure 7b depicts the packets sent in the normal simulation using a bar chart. Similar to Figure 6a,b, the “CAN_ID” on the *x*-axis signifies the message IDs in each simulation, while “count” on the y-axis denotes the total packets sent by each message ID.

In Figure 7a, the bar displaying the highest count value is the compromised message ID (0x00), indicating the extensive control of the attacker over the bus to prevent the transmission of signals from legitimate message IDs within the system. The attacker leveraged the arbitration feature of the CAN bus by flooding the bus with signals from a message ID 0x000, thereby ensuring it wins the arbitration process. The staggering difference between the highest total number of packets sent and the subsequent packets indicates the DoS attack. On the other hand, Figure 7b shows the typical variations of packets sent by each node when the system operates in its standard mode.

### 5.3. Test Run 3 Result

In the final cycle, we monitored the system after an hour in both the normal and attack scenarios. Analogous to Test Runs 1 and 2, we also monitored the traffic sent and logged the data using a sniffer, eventually storing the raw data in CSV files. We discuss our findings after processing the data using Python below:Attack simulation: In this cycle, the compromised node sent over 6112 packets, which is over two-times the total packets sent in Test Run 2. The following message ID sent only 642 packets, an astounding percentage decrease of 852.96% compared to the highest number of packets recorded for message ID 0x00. The major difference between the total packets sent by the compromised message ID and other message IDs in the simulation shows that the attacker had major control of the bus, thereby causing a DoS attack. We represent the total number of packets sent in this phase in Table 9.Normal simulation: In this simulation, like in the prior test runs discussed, we observed a fair access of each message ID to transfer signals via the bus by the uniform distribution of packets across the system. The highest number of packets sent was 120, from message ID 0000006D. The slight difference between the total number of packets recorded for each message ID in this phase indicates the absence of abnormal activities in this phase, as each message ID had a fair opportunity to transmit signals within the system. In Table 10, we present the total number of packets sent by each message ID within the 1 h time frame.

We visually represent the tabular data presented in Table 9 and Table 10 using bar charts to observe the variations of packet transfer across all message IDs. Figure 8a shows the bar chart of the total number of packets sent in the attack simulation displayed in Table 9. In this figure, we see a spike for the message ID 0x000, indicating the continuous flooding of packets from this message ID on the bus during the time of monitoring. Figure 8b shows the bar chart of the total packets sent in the normal simulation after 1 h.

After all three runs, our experimentation showed that the CAN protocol is prone to various security vulnerabilities, particularly DoS, spoofing, and injection. In all three runs, the compromises ID “0000000” sent increasingly high numbers of packets, which could be detrimental to the CAN bus primarily because it prevents the system from obtaining signals from legitimate ECUs by flooding irrelevant signals.

### 5.4. Latency of the CAN Bus System

Here, we examine the latency of the CAN bus system when it is under attack. Understanding CAN bus latency is critical for creating real-time systems that rely on timely and coordinated communication between nodes [43]. Latency reduction is vital for ensuring efficient and dependable performance.

Latency in the CAN bus is the total time it takes for a message to propagate across the bus from when a node initiates it until another node receives it [44]. In our proposed attack scenario, it is important to note that the attacker’s node is stationary. This static configuration can indeed have an impact on message latency. Furthermore, factors such as propagation delay, bit rate, bus length, transmission delay, and arbitration time can also contribute to variations in CAN bus latency [45]. Consequently, calculating the precise latency of the bus in such a complex system can be challenging. However, in our research, we made an effort to examine the latency of the CAN simulation system under the three test runs detailed in Section 4.

To calculate the latency, we used a mathematical formula shown in Equation (Equation 1). This formula focuses on the time delay for message transfer from the same node on the bus.
(1)$$\Delta x=x_{i+1}-x_{i},$$
where:$x_{i}$ = timestamp for the current message transmitted on the bus from a specific ID.$x_{i+1}$ = timestamp for the preceding message transmission from the same message ID.Δx = latency.

In our analysis, we took the difference between these variables as the system’s latency. The timestamp in this research was measured in milliseconds (ms); latency was also measured in ms. We examined the latency of the system in each of the three attack implementation cycles explained in Section 4 and visualize our results using Kernel Density Estimation (KDE) graphs.

KDE is a statistical approach for estimating the probability density function of a continuous random variable based on observed data. In KDE, we constructed a kernel (usually a smooth, bell-shaped function) at each data point and added these kernels to generate a continuous probability density curve. This curve depicts the probability of the observation of data values at various positions along the *x*-axis. Upon viewing the graphs illustrating the latency values in Section 5.4.1, Section 5.4.2 and Section 5.4.3, we notice some negative values in the *x*-axis. Negative *x*-axis values do not signal the presence of negative data values in the dataset. They, instead, highlight areas where the predicted probability density is lower than the general average. In other words, these negative values show portions of the distribution where data points are less likely to occur than in the center. The presence of negative values in KDE is a natural result of the method’s capacity to capture fluctuations in data density over the variable’s entire range. It is critical to understand these negative values within the context of probability density estimates, viewing them as indicators of lower probability density regions rather than true negative data values. We explain the analysis of the latency in all three test runs previously discussed in Section 4 below.

#### 5.4.1. Latency of the System in Test Run 1

We monitored the signals sent within 15 min of this attack simulation. The signals were stored in a CSV file and processed using Python. In the file containing the raw data captured, two columns were significant for calculating the latency: the timestamp column and the message ID column. We organized the data to depict the difference between the time a message ID transmits a signal and the time that same message ID transmits its next message.

During the attack simulation in this cycle, a notable observation was the consistent value of latency recorded for the packets sent from CAN ID “00000000”. The latency value for every packet sent from this particular ID was recorded as 3 ms. Furthermore, a unique observation was made that following the compromised ID “00000000”, the preceding subsequent message exhibited a significant spike in latency. All the highest latencies were recorded after a node transmitted after ID “00000000”. In Figure 9, we show the frequency of the latency distribution for data captured in the first attack cycle. The frequency values show the count per millisecond of each latency value calculated. The blue line in Figure 9a represents the frequency of the same latency value (3 ms) recorded for the ID “0x000”. From this figure, we see a spike for the ID “0000000”, indicating a higher frequency for the same latency value (3 ms) from that particular message ID. This implies a consistent pattern of signal transmission from the attacker.

In retrospect, upon reviewing the total packets sent in the attack simulations discussed in Section 5, we can correlate the high packet transfer with the latency value displayed in Figure 9a. This correlation suggests that the high packet transfer rate aligns with the repeated occurrence of the 3 ms latency value. Consequently, the preceding message IDs recorded extremely low frequencies for the latency distribution of signals sent on the bus. This indicates varying time space for message transfer across the bus, ultimately signaling the long wait time for legitimate nodes to transfer data on the bus. Furthermore, to offer a more-comprehensive representation of the latency variations for the legitimate ECUs with non-zero IDs, we created Figure 9b, which focuses solely on the legitimate IDs, excluding the illustration of the attacker’s ID. This approach allowed us to expand the depiction of latency variations specifically for the legitimate ECUs.

Figure 10 illustrates the frequency of the latency distribution for the normal simulation after 15 min. In this figure, we note a relatively uniform distribution in the increase and decrease of the latency across the bus. Upon reviewing the x-axis in Figure 10, the latency is recorded in milliseconds, and the higher frequency for almost all message IDs’ latency is situated between 500 ms and 10,000 ms. This indicates similar wait times across the bus for the message IDs. The impact of this is that it shows no abnormal behavior regarding the systems’ performance analysis.

#### 5.4.2. Latency of the System in Test Run 2

In Figure 11, we show the frequency of the latency distribution for the attack simulation in Test Run 2. We calculated the latency of all the packets sent after allowing this simulation to run for 30 min. Notably, the lowest recorded latency was 7 ms, and all of these packets originated from the ID “000000000”. Furthermore, we observed a consistent pattern, where after the bus processes packets sent from the ID “00000000”, there is an increased time delay recorded for the subsequent packet transmitted via the bus. In Figure 11a, the blue line signifies the frequency of the latency for the message ID “0000000”. We see this value as a straight line because the same value was recorded for that message ID, and this value was 7 ms. That is, after every 7 ms, the attacker sent a message to the bus. Upon correlating the total packets sent in this simulation as discussed in Section 5, in 30 min, the attacker sent 3556 packets, with each packet sent every 7 ms.

A major significance of reviewing the latency distribution is to monitor the timing of the message transfer within the system. From examining Figure 11a, an abnormal activity we can see reoccurring is the consistent value for latency from the same message ID. This lets us know that the system is behaving in a suspicious manner. Additionally, in the context of this simulation, the frequencies of other message IDs are depicted as straight lines positioned at the bottom of the graph. This positioning suggests that almost all legitimate message IDs experience wait times that can be as long as 50,000 ms. This further buttresses that the CAN bus is highly prone to DoS attacks as the high latency values for the latency of legitimate message IDs signify delays in the system’s real-time response and performance.

Furthermore, to provide a more-comprehensive representation of latency variations for the legitimate ECUs with non-zero IDs, we introduce Figure 11b, which specifically focuses on the legitimate IDs. In this figure, the illustration of the attacker’s ID has been excluded, allowing us to expand the depiction of latency variations for the legitimate ECUs. We positioned this figure side by side to visualize the impact of the attacker.

In Figure 12, the frequency of the latency distribution for the normal simulation in Test Run 2 is shown. Here, we observe that the latency increases steadily until the bus goes idle. The lowest latency recorded in this observation was 29 ms. This is normal as the CAN bus is more centered towards fault tolerance than low latency.

Figure 12 also shows that all message IDs display similar values for the frequency of the latency, indicating the absence of abnormal activities such as unexpected high latency values, which could cause delays or consistently small latency values, which could prevent the transfer of signals from other message IDs. The impact of the similarity in the gradients of the frequency for the latency distribution indicates that the system is working in typical operations.

#### 5.4.3. Latency of the System in Test Run 3

In Figure 13a,b, we show the latency distribution for the attack simulation of Test Run 3. This simulation revealed a similar occurrence when compared to the previous two attack simulations that were studied. Notably, we recorded the lowest latency distribution from message ID “0000000” at 7 ms.

Similar to Figure 9a and Figure 11a, we notice that the “blue line” in Figure 13a signifies the frequency of the latency values associated with the message ID “0000000”. The graph shows a straight line for this value (7 ms) because the same latency value is consistently recorded for that specific message ID. The consistent latency pattern is ominous as it prevents other message IDs from sending legitimate signals to the system and continuously floods the system with irrelevant signals every 7 ms. The impact of this is the poor performance of the system, which could be detrimental to the vehicles’ overall performance, thereby leading to dire effects that could compromise safety.

Further, analogous to Figure 9b and Figure 11b, to offer a more-comprehensive view of the latency variations among the legitimate ECUs with non-zero IDs, we introduce Figure 13b, concentrating solely on these legitimate IDs. In this figure, we deliberately excluded the depiction of the attacker’s ID, enabling us to enhance the visualization of latency variations among the legitimate ECUs. In the case of Figure 13b, we arranged this figure side by side with Figure 13a to illustrate the influence of the attacker in the third attack scenario.

Figure 14 displays the latency frequency in our 60 min normal simulation, illustrating message transfer timing in regular operations. Figure 14, akin to Figure 8 and Figure 12, exhibits consistent timing gradients for message transfer across all message IDs on the bus, primarily because they operate in normal simulation. In this figure, we understand the timing behavior of the network, and it shows proper communication and synchronization between nodes transmitting messages. Consequently, Figure 14 leads to the inference that all message IDs transmitted messages devoid of any irregularities.

#### 5.4.4. Average of the Latency in All Three Test Runs

We calculated the average latency of both normal and attack scenarios studied in this research. This is because we believe it is useful for a variety of research and analytical applications as it provides a summary metric that may be used to evaluate the overall performance of the CAN system. This indicator allows us to effectively monitor how the system performs on average, providing significant insights into its efficiency and dependability.

Furthermore, computing the average latency can serve as a valuable tool for trend analysis. By tracking changes in the average latency over time, it will be easier to identify patterns or trends. These trends may include gradual degradation of performance or improvements resulting from system updates or optimizations. Trend analysis enhances the ability to fine-tune systems for optimal performance and security.

Therefore, to calculate the average of the latency values in this research, we used the formula below:(2)$$\mathrm{Average}=\frac{\sum_{i=1}^{n}x_{i}}{n}.$$
where:$x_{i}$ = represents each individual latency value in that particular simulation*n* = the total number of packets sentAverage = average of the latency for that particular simulation.

We display the average of the latency values for all three test runs in Table 11 below.

In Table 12, we summarize the latency distributions for all experimental runs in this research.

From our analysis of the average in regards to the latency values in Table 11, we can conclude that, when the average latency for normal messages exceeds that of attack messages in a CAN-enabled system, it may suggest that normal traffic within the system may inherently exhibit higher latency due to factors such as the message volume, priority, or system design. Conversely, a lower average latency for attack messages may signify attackers’ attempts to efficiently disrupt the network or execute attacks with minimal latency. This discrepancy can also serve as an effective anomaly-detection mechanism, flagging potential issues or security breaches when deviations occur.

## 6. Conclusions

In this research, we conducted a security assessment on a simulation of a CAN-enabled system, building upon our previous work described in [1]. We proposed an attack scenario and implemented it in three cycles to analyze the evolving effect of the attack over time. Through a comparative analysis, we evaluated the system’s performance under normal and attack operations. Additionally, we examined the latency within the system to assess the time delay between transmitted messages via the bus.

The analysis revealed vulnerabilities within the CAN bus that render it susceptible to some attacks, such as injection, spoofing, and DoS attacks. Our analysis revealed a major vulnerability of the CAN bus to DoS attacks. The results obtained from the three monitored simulations demonstrated that, when an attacker successfully spoofs the message ID to “0000000” within a vehicle, the attacker gains control over the bus, allowing the attacker to flood it with a high volume of packets using extremely low latency times. This malicious activity effectively prevents authorized nodes from accessing the bus when required. The consistently low latencies observed from the same message ID serve as a strong indicator of a potential attack on the bus. In normal operations, we observed that the bus prioritizes reliability over low latency. Therefore, the presence of such low latencies suggests that the bus might be under attack, mainly because relatively low latencies are not common when the CAN bus is not under attack. These findings highlight the importance of conducting such investigations to identify vulnerable areas that malicious attackers may exploit for personal gain. Ultimately, in this paper, we presented the vulnerabilities we uncovered in the CAN bus, including the potential for some attacks such as spoofing, injection, and DoS. We also highlighted the role of abnormally low latencies as an indicator of suspicious activity. By addressing these issues, we can strengthen the security and reliability of CAN-enabled systems. In the future, our goal is to develop an effective system that monitors and halts attacks within the bus.

## Figures and Tables

**Figure 1 sensors-23-08223-f001:**
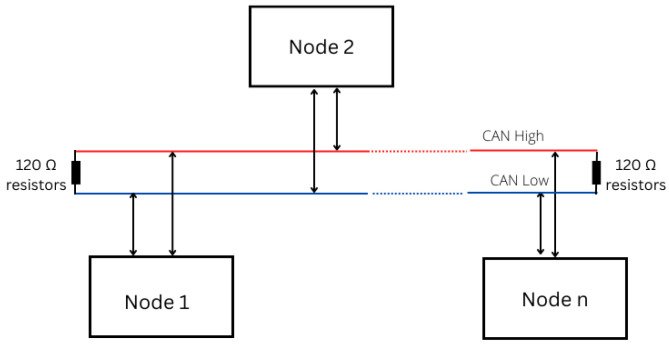
The CAN network [1]: This illustrates a CAN-enabled network with n-number of nodes sending messages via the bus for the vehicle’s overall functionality. The nodes can be the engine, radio, brake, steering wheel, etc.

**Figure 2 sensors-23-08223-f002:**
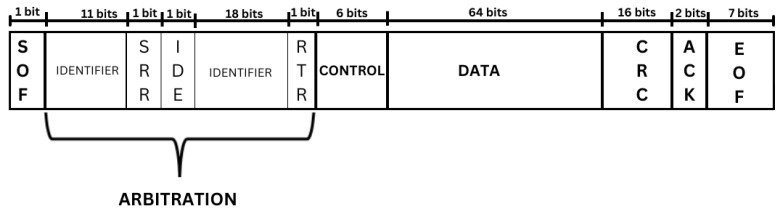
The frame format for an extended CAN message: This shows an example of a message packet sent via the bus. When the message is decoded, we can retrieve all of its components based on the size of each field.

**Figure 3 sensors-23-08223-f003:**
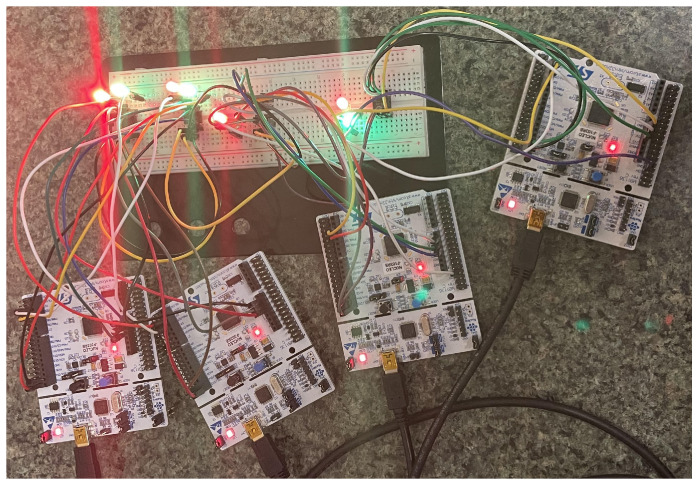
The implementation of a testbed for a CAN bus system [1]: A testbed of a CAN-enabled system with four nodes attached to it. These four nodes communicate via the CAN high and low wires.

**Figure 4 sensors-23-08223-f004:**
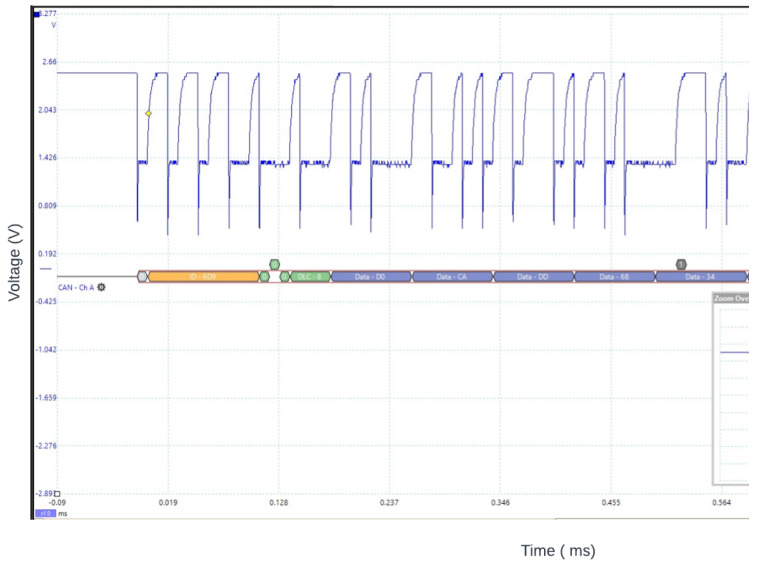
A CAN message decoded into its individuals assembling fields using the PicoScope [1]: An actual representation of a real CAN message packet decoded into its various fields, as illustrated in Figure 2.

**Figure 5 sensors-23-08223-f005:**
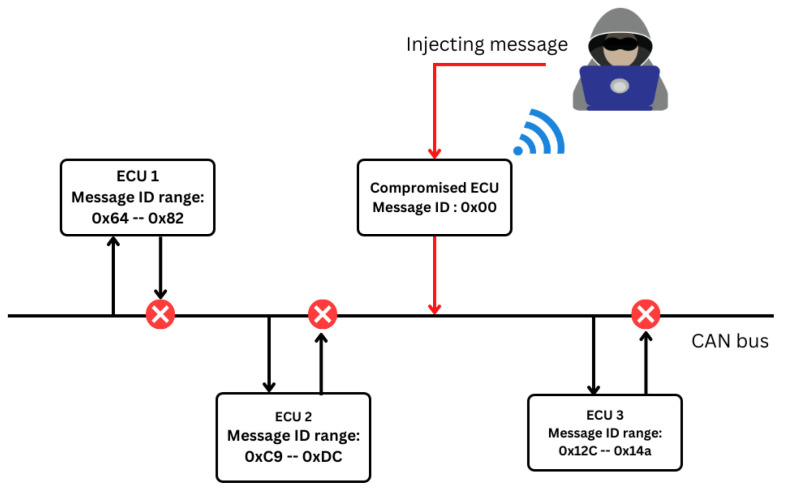
Proposed attack design scenario: An ECU is hijacked by an attacker, and the message ID is spoofed to hex 0x00, ensuring that messages from that ECU/node always win the arbitration process, thereby allowing for continuous injection of the signals and, consequently, preventing the transmission of other messages from legitimate nodes on the bus (DoS).

**Figure 6 sensors-23-08223-f006:**
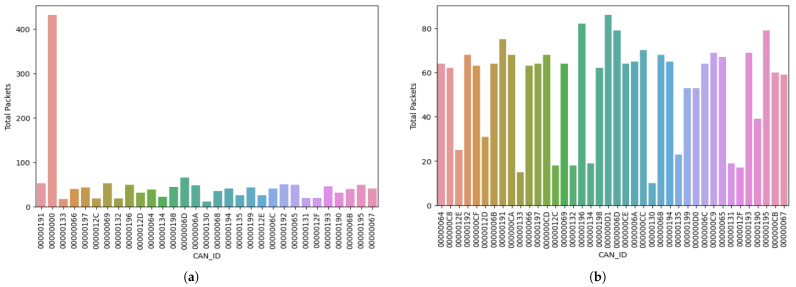
Test Run 1: Bar chart of the total packets sent in the attack and normal simulations after 15 min. (**a**) Bar chart of the total packets sent in the attack simulation after 15 min. (**b**) Bar chart of the total packets sent in the normal simulation after 15 min.

**Figure 7 sensors-23-08223-f007:**
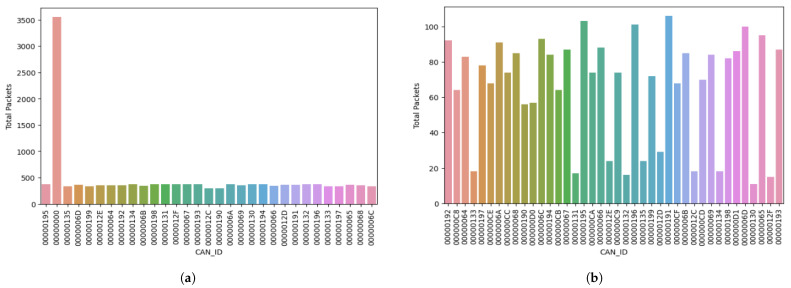
Test Run 2: Bar chart of the total packets sent in the attack and normal simulations after 30 min. (**a**) Bar chart of the total packets sent in the attack simulation after 30 min. (**b**) Bar chart of the total packets sent in the normal simulation after 30 min.

**Figure 8 sensors-23-08223-f008:**
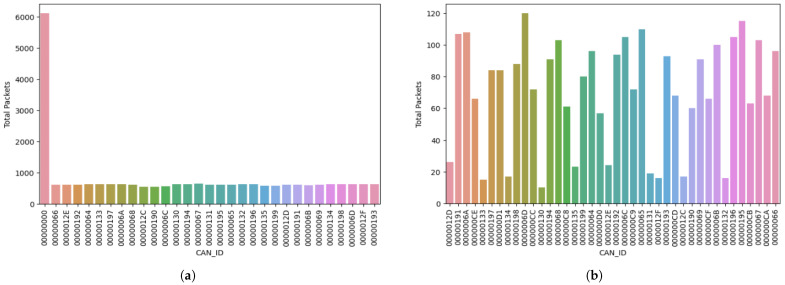
Test Run 3: Bar chart of the total packets sent in the attack and normal simulations after 60 min. (**a**) Bar chart of the total packets sent in the attack simulation after 60 min. (**b**) Bar chart of the total packets sent in the normal simulation after 60 min.

**Figure 9 sensors-23-08223-f009:**
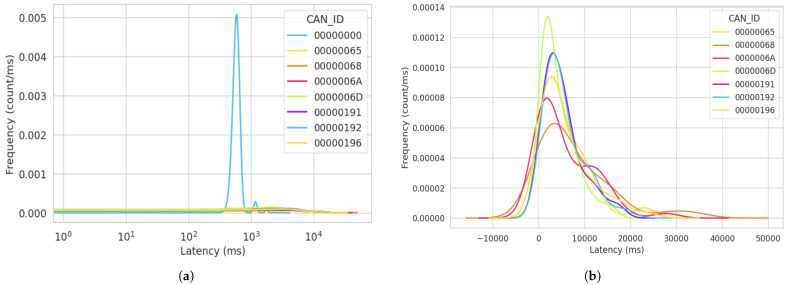
Frequency of latency distribution for the first attack simulation after 15 min. (**a**) KDE log plot of the latency distribution for the first attack simulation. (**b**) Latency variations for the non-zero CAN ID in the first attack simulation.

**Figure 10 sensors-23-08223-f010:**
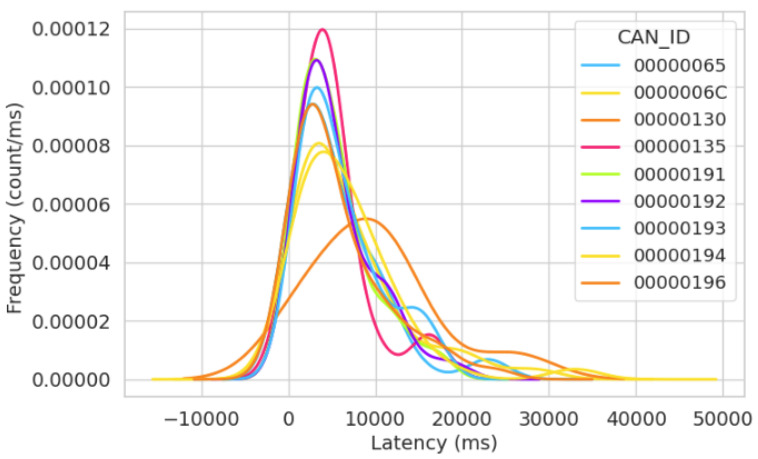
Frequency of latency distribution for the first normal simulation after 15 min.

**Figure 11 sensors-23-08223-f011:**
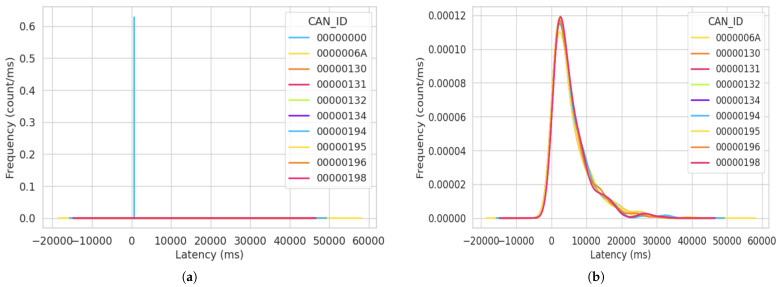
Frequency of latency distribution for the second attack simulation after 30 min. (**a**) KDE latency distribution for the second attack simulation. (**b**) Latency variations for the non-zero CAN ID in the second attack simulation.

**Figure 12 sensors-23-08223-f012:**
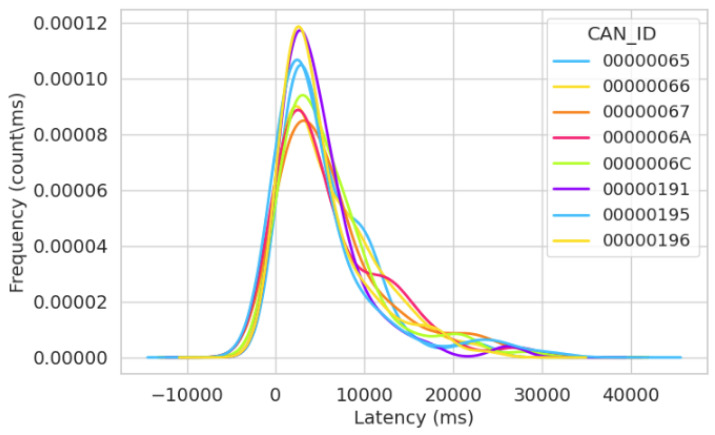
Frequency of latency distribution for the second normal simulation after 30 min.

**Figure 13 sensors-23-08223-f013:**
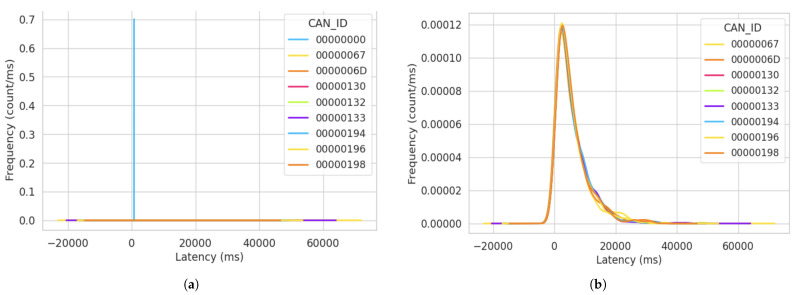
Frequency of latency distribution for the third attack simulation after 60 min. (**a**) KDE latency distribution for the third attack simulation. (**b**) Latency variations for the non-zero CAN ID in the third attack simulation.

**Figure 14 sensors-23-08223-f014:**
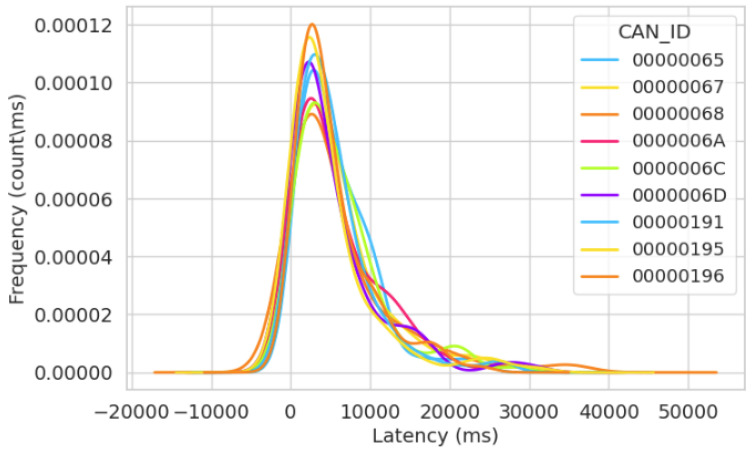
Frequency of latency distribution for the third normal simulation after 60 min.

**Table 1 sensors-23-08223-t001:** Hardware used to create the CAN bus testbed.

Hardware	Purpose
STM Nucleuo– F103RB	The Microcontroller (MCU) used for the research.
MCP 2551	Transceivers to allow for the CAN communication on the
CL2000	This device was used to log CAN traffic from the bus (i.e., the sniffer). It included the timestamps, data length, bus rate, and message IDs.
PicoScope 2204A	An oscilloscope used to decode the traffic sniffed from the bus.

**Table 2 sensors-23-08223-t002:** ECUs and Message IDS in the proposed attack scenario.

ECU	Number of Assigned Message ID	Message ID (Hexadecimal Notations)
ECU 1	10	0x64 – 0x82
ECU 2	10	0xC9 – 0xDC
ECU 3	10	0x12C – 0x14A
Compromised ECU	1	0x00

**Table 3 sensors-23-08223-t003:** Implementation cycles and their time intervals.

Implementation Cycles	Time Interval
Test Run 1	15 min
Test Run 2	30 min
Test Run 3	60 min

**Table 4 sensors-23-08223-t004:** Simulations and their descriptions.

Perspective	Description
Normal Simulation	This represents the operation where all four nodes are working properly without any attack. In this case, we assigned ten message IDs to each node.
Attack Simulation	In this scenario analysis, we examined the performance of a system operating during the attack scenario discussed earlier. That is, one compromised node continuously injected messages into the bus with ID 0x000.

**Table 5 sensors-23-08223-t005:** Test Run 1 (attack simulation) total packets.

Message ID	Packets Sent in 15 min
00000000	432
0000006D	66
00000191	53
00000069	52
00000192	50
00000195	49
00000065	49
00000196	49
0000006A	48
00000193	46
00000197	43
00000199	43
0000006C	41
00000194	41
00000067	41
0000006B	40
00000066	39
00000064	38

**Table 6 sensors-23-08223-t006:** Test Run 1 (normal simulation) total packets.

Message ID	Packets Sent in 15 min
000000D1	86
00000196	82
0000006D	79
00000195	79
00000191	75
000000CC	70
00000193	69
000000C9	69
00000192	68
00000068	68
000000CD	68
000000CA	68
00000065	67
00000194	65
0000006A	65
0000006C	64
000000CE	64
00000064	64

**Table 7 sensors-23-08223-t007:** Test Run 2 (attack simulation) total packets.

Message ID	Packets Sent in 30 min
00000000	3556
00000130	377
00000134	376
00000194	376
00000198	376
00000196	374
00000132	374
00000195	373
0000006A	372
00000131	372
00000067	372
0000012F	370
00000193	370
0000006D	365
0000012D	364
00000191	364
00000065	361
00000068	354
00000064	352
0000012E	352
00000069	352
00000192	352
00000066	349
0000006B	346
00000133	339

**Table 8 sensors-23-08223-t008:** Test Run 2 (normal simulation) total packets.

Message ID	Packets Sent in 30 min
00000191	106
00000195	103
00000196	101
0000006D	100
00000065	95
0000006C	93
00000192	92
0000006A	91
00000066	88
00000067	87
00000193	87
000000D1	86
0000006B	85
00000068	85
00000194	84
00000069	84
00000064	83
00000198	82
00000197	78
000000C9	74
000000CC	74
000000CA	74
00000199	72
000000CD	70
000000CE	68

**Table 9 sensors-23-08223-t009:** Test Run 3 (attack simulation) total packets.

Message ID	Packets Sent in 60 min
00000000	6112
00000067	642
00000196	635
00000132	635
0000006D	632
00000130	631
00000194	631
00000133	626
00000198	625
00000197	625
00000134	625
0000012F	624
00000193	624
0000006A	623
00000064	622
0000012E	616
00000192	616
00000069	615
0000012D	614
00000191	614
00000066	613
00000131	611
00000195	611

**Table 10 sensors-23-08223-t010:** Test Run 3 (normal simulation) total packets.

Message ID	Packets Sent in 60 min
0000006D	120
00000195	115
00000065	110
0000006A	108
00000191	107
00000196	105
0000006C	105
00000068	103
00000067	103
0000006B	100
00000066	96
00000064	96
00000192	94
00000193	93
00000069	91
00000194	91
00000198	88
00000197	84
000000D1	84
00000199	80
000000CC	72
000000C9	72
000000CA	68

**Table 11 sensors-23-08223-t011:** Average of the latency values in all three test runs.

Simulation	Average for Normal Simulation (ms)	Average for Attack Simulation (ms)
Test Run 1	189.137	183.124
Test Run 2	210.080	147.188
Test Run 3	225.965	147.189

**Table 12 sensors-23-08223-t012:** Summary of latency analysis.

Implementation Cycle	Simulation	Observations
Test Run 1	Attack Simulation	Lowest Latency recorded 3 ms. The CAN ID “00000000” recorded the lowest latency. Nodes that send messages after the compromised CAN ID recorded the highest latency.
	Normal Simulation	This had a more-uniform distribution. The lowest latency recorded was 9 ms.
Test Run 2	Attack Simulation	Lowest latency recorded was 7 ms The ID “00000000” always recorded the lowest latency. Increased time delay recorded for the subsequent packet transmitted via the bus.
	Normal Simulation	Steady increase in latency until the bus becomes idle. The lowest latency recorded was 29 ms.
Test Run 3	Attack Simulation	Lowest latency of the compromised CAN ID “000000000” was 7 ms. This value was consistent for all signals sent from this ID. Signals sent after this compromised ID recorded the lowest latency at 8 ms.
	Normal Simulation	Lowest latency recorded was 12 ms. The messages were distributed fairly across the bus. More message IDs were recorded in this time phase.

## Data Availability

Not applicable.

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
