# Peer review of "CANAttack: Assessing Vulnerabilities within Controller Area Network"

_sensors, 2023, doi:10.3390/s23198223_

Round 1
Reviewer 1 Report
Dear Authors
The paper is drafted well; however there are many relevant works that has been carried out, the paper can be improved by adding more relevant research.
None
Author Response
Thank you for reviewing our paper “sensors-2630220”. We appreciate the time taken for your valuable feedback that has been provided. We have considered the feedback and revised our manuscript accordingly. Our responses are detailed in this attached letter.

Reviewer 2 Report
First of all, I would like to congratulate the authors on undertaking an interesting research topic. Everything related to security is an important aspect of research. In the introduction, the authors presented current reports related to the research topic. In my opinion, it was presented correctly. The research methodology was clearly presented.
Fig. 7b, Fig.9 -14. suggests adding a unit to the Y axis (frequency) as on the x axis.
In my opinion, the conclusions provide a general summary of the research.
The literature contains 24 items, most of which are the latest reports.
Only items 5 and 25 constitute the theoretical basis presented long ago.
In my opinion, the article is suitable for publication in the journal Sensors in Special Issue.
Author Response

(The authors gave the same response as above.)

Reviewer 3 Report
This paper presents a security vulnerability on CAN bus network that utilizes CAN messages arbitration mechanism to launch a Denial of Service (DoS) on CAN bus. The authors discussed the introduction of a new testbed that makes use of multiple embedded devices with CAN bus capability to implement and test the impact of the proposed threat model. Test performance based on latency and number of injected packets for the proposed threat model were fully presented in this research.
However, there are few shortcomings in the paper that the authors need to address to further improve this paper. These are listed below.
1- The authors have discussed various threat models in the paper such spoofing and injection attacks. I was unable to determine if these two threats were ever implemented in this paper? If yes, the authors need to elaborate on them more .
2- The author has mentioned that CAN data frames are generated from several ECU nodes attached to real CAN bus network. I was unable to understand how random CAN ID were injected into the bus. The authors need to discuss how random CAN messages are generated and injected into the bus.
3- Based on the system architecture, there is one ECU node acting as an attacker. Is this node stationary? Or the attack can run on different ECUs attached to the CAN bus. I believe the physical location of the infected ECU will have a direct impact on messages latency. I think the authors need to address this point in the paper.
4- In Figures 9 - 14 , the x axis has negative values but the measurements is time. The authors need to correct this mistake in paper.
5- In the latency figures, figure 9, 11, and 13, packet IDs with non zero value
shows a flat line. These figures need to be expanded to show latency variations for the non zero CAN ID.
6- The authors need also to present the message average latency for both normal and malicious CAN messages.
NA
Author Response
hank you for reviewing our paper “sensors-2630220”. We appreciate the time taken by each of the reviewers and the valuable feedback that has been provided. We have considered the feedback and revised our manuscript accordingly. Our responses are detailed in the attached letter
